# AP Collagen Peptides Prevent Cortisol-Induced Decrease of Collagen Type I in Human Dermal Fibroblasts

**DOI:** 10.3390/ijms22094788

**Published:** 2021-04-30

**Authors:** Minjung Chae, Il-Hong Bae, Sunghwan Lim, Kyoungmi Jung, Jonghwa Roh, Wangi Kim

**Affiliations:** Amorepacific Research and Development 1, 314-1, Bora-dong, Giheung-gu, Yongin-si 446-729, Gyeonggi-do, Korea; baelong@amorepacific.com (I.-H.B.); limsungh@amorepacific.com (S.L.); jkm@amorepacific.com (K.J.); rohjh@amorepacific.com (J.R.); katemina@amorepacific.com (W.K.)

**Keywords:** glucocorticoid, cortisol, glucocorticoid receptor, GR, fish collagen, procollagen, TGF-β

## Abstract

Cortisol is an endogenous glucocorticoid (GC) and primary stress hormone that regulates a wide range of stress responses in humans. The adverse effects of cortisol on the skin have been extensively documented but the underlying mechanism of cortisol-induced signaling is still unclear. In the present study, we investigate the effect of cortisol on collagen type I expression and the effect of AP collagen peptides, collagen tripeptide-rich hydrolysates containing 3% glycine-proline- hydroxyproline (Gly-Pro-Hyp, GPH) from the fish skin, on the cortisol-mediated inhibition of collagen type I and the cortisol-induced signaling that regulates collagen type I production in human dermal fibroblasts (HDFs). We determine that cortisol downregulates the expression of collagen type I. AP collagen peptides or GC receptor (GR) inhibitors recover the cortisol-mediated inhibition of collagen type I and GR activation. AP collagen peptides or GR inhibitors also prevent the cortisol-dependent inhibition of transforming growth factor (TGF)-β signaling. AP collagen peptides or GR inhibitors are effective in the prevention of collagen type I inhibition mediated by cortisol in senescent HDFs and reconstituted human skin models. Taken together, GR signaling might be responsible for the cortisol-mediated inhibition of TGF-β. AP collagen peptides act as GR-mediated signaling blockers, preventing the cortisol-dependent inhibition of collagen type I. Therefore, AP collagen peptides have the potential to improve skin health.

## 1. Introduction

Glucocorticoids (GCs), cortisol in humans or corticosterone in rodents, are steroid hormones predominantly produced in the adrenal glands in response to various stressors such as physical injury and psychological stress [1]. In humans, the hypothalamus-pituitary-adrenal (HPA) axis coordinates the synthesis of adrenal GCs. GCs exert their effects by binding to the glucocorticoid receptor (GR) and the mineralocorticoid receptor (MR), which are structurally and functionally homologous ligand-activated transcription factors (TFs) that belong to members of the nuclear receptor (NR) superfamily [2,3,4,5]. GR is ubiquitously expressed, while MR has a more restricted expression pattern with the highest levels found in the kidney [6,7]. Both receptors are expressed in the dermis [8]. In response to endogenous hormones or synthetic ligands, GR and MR dissociate from multimeric cytoplasmic inhibitory complexes, undergo post-translational modification, translocate to the nucleus and bind to GC response elements (GREs) in target genes, or directly interact with transcription factors, such as activating protein-1 (AP-1) and nuclear factor-κB (NF-κB) [9,10].

The skin is the outermost barrier of the body and maintains homeostasis between the external environment and internal tissues. Stress can have detrimental effects on skin integrity and function. Psychological stress is known to exacerbate features of skin diseases such as psoriasis and atopic dermatitis through the release of cortisol, which regulates a wide range of stress responses [11]. Chronic GC excess, either due to exogenous administration or endogenous deregulation, as in Cushing’s syndrome, causes skin alterations, such as atrophy and impaired wound healing [12,13]. These abnormalities are very similar to those observed with aging [14,15]. Thus, the therapeutic use of cortisol is limited due to the adverse side-effects, even if cortisol is one of the most anti-inflammatory compounds for treating inflammatory and autoimmune diseases.

Collagen is the main extracellular matrix (ECM) protein in animal tissues and the most abundant protein in the body, comprising approximately one-third of total protein. Collagen has a unique triple-helix structure with a repeated amino acid sequence of (Gly-X-Y)_n_, in which the most frequently occurring sequence is Glycine-Proline-Hydroxyproline (Gly-Pro-Hyp, GPH) [16]. Collagen hydrolysates intended for food supplements are produced from collagen extracted from animals or fish by enzymatic digestion. Over recent years, the consumption of collagen hydrolysates has been reported to improve skin moisture, elasticity, wrinkles [17,18] and barrier function [19].

In this study, we evaluate the decrease in collagen type I expression in cortisol-stimulated human dermal fibroblasts (HDFs). Inhibition of collagen type I is mediated by GR activation. Cortisol-mediated inhibition of the transforming growth factor (TGF)-β signaling regulating collagen deposition is dependent on GR activation. We also test the effect of AP collagen peptides with a high level of tripeptides including 3% GPH, produced by the enzymatic hydrolysis of collagen derived from the skin of golden threadfin bream, on collagen type I. AP collagen peptides prevent the cortisol-induced decrease in collagen type I via GR inactivation. AP collagen peptides are also effective in preventing the cortisol-dependent inhibition of collagen type I in replicative senescent HDFs and reconstituted skin models. Therefore, AP collagen peptides play a significant role in collagen synthesis, highlighting their potential application as functional foods to combat cortisol/aging-induced impairment of dermal integrity.

## 2. Results

### 2.1. Cortisol Inhibits Collagen Type I Synthesis in Human Dermal Fibroblasts

We investigated the effects of cortisol treatment on collagen type I synthesis in human dermal fibroblasts (HDFs). The gene and protein expression levels of collagen type I were measured after treatment with three different concentrations of cortisol (1, 10 and 100 nM). Cell viability was not affected by these treatments (Figure 1A). We analyzed the transcript levels of col1a1 using quantitative real-time PCR and found that cortisol decreased the mRNA expression of col1a1 in a dose-dependent manner (Figure 1B). To confirm the inhibitory effect of cortisol on collagen type I expression, the supernatant of the cells was analyzed for the detection of protein levels. Cortisol treatment resulted in a dose-dependent reduction in the protein expression levels of procollagen type I (Figure 1C). These data suggest that cortisol inhibits collagen type I synthesis in HDFs.

### 2.2. AP Collagen Peptides and GPH Prevent the Inhibition of Collagen Type I Mediated by Cortisol

Ingestion of collagen hydrolysates enhances collagen synthesis in the skin [18]. To elucidate the effect of AP collagen hydrolysates on collagen synthesis, HDFs were treated with cortisol in the presence of two different concentrations of AP collagen peptides (0.1 and 1 μg/mL). We analyzed the transcript levels of col1a1 in cortisol-stimulated HDFs via quantitative real-time PCR and found that AP collagen peptides recovered the cortisol-mediated inhibition of col1a1 mRNA expression in a dose-dependent manner (Figure 2A). To confirm the effect of AP collagen peptides on collagen production, the protein levels of procollagen type I were determined in the supernatants of HDFs. Our results showed that AP collagen peptides restored the cortisol-dependent reduction of procollagen type I in a dose-dependent manner (Figure 2B). Taken together, AP collagen peptides prevented the cortisol-mediated downregulation of collagen type I.

GPH is one of the major tripeptides of AP collagen peptides, accounting for approximately 3% of the total components. To explore the role of GPH in collagen synthesis, we analyzed the gene and protein levels of collagen type I in HDFs treated with cortisol in the presence of two different concentrations of GPH (0.01 and 0.03 μg/mL). The transcript levels of col1a1 indicated that GPH recovered the cortisol-mediated inhibition of col1a1 mRNA expression in a dose-dependent manner (Figure 2C). We also observed that GPH recovered the inhibition of procollagen type I secretion mediated by cortisol (Figure 2D). Collectively, the cortisol-dependent inhibition of collagen type I synthesis is prevented by AP collagen peptides, which may be partially attributed to GPH.

### 2.3. GR Signaling Is Responsible for Cortisol-Dependent Inhibition of Collagen Type I Synthesis and AP Collagen Peptides Inhibit GR Activation

While both GR and MR are present in all skin structures, the specific roles of MR and GR in the skin remain under investigation [20]. To determine which receptor signaling is responsible for the inhibition of collagen type I synthesis, HDFs treated with cortisol in the presence of either GR antagonist RU486 or MR antagonist spironolactone were analyzed for the gene and protein levels of collagen type I. GR antagonist stimulated the synthesis of col1a1 mRNA, which was inhibited by cortisol treatment (Figure 3A). Prevention of cortisol-dependent downregulation of procollagen type 1 secretion was also observed with RU486 treatment (Figure 3B). These data suggest that GR signaling activated by cortisol leads to a decrease in collagen type I synthesis.

GR regulates a wide range of metabolic and developmental processes by phosphorylation at multiple serine residues in a hormone-dependent manner [21,22]. Ser211 phosphorylation, a biomarker for activated GR, is induced by cortisol [23,24]. To elucidate the role of AP collagen peptides in GR activity, GR phosphorylation was assessed using phosphor-Ser211 antibodies. Our results demonstrated that cortisol stimulated GR phosphorylation and that AP collagen peptides or RU486 attenuated cortisol-induced GR phosphorylation (Figure 3C,D). Taken together, GR signaling is responsible for the cortisol-dependent inhibition of collagen type I synthesis, and AP collagen peptides may act as inhibitors of GR phosphorylation.

### 2.4. AP Collagen Peptides Recover Cortisol-Dependent Inhibition of TGF-β Signaling

TGF-β, a major regulator of ECM metabolism, [25] regulates collagen homeostasis by collagen production and degradation via the Smad pathway [26]. TGF-β binds to its receptor complex, thereby stimulating intracellular Smad 2/3 phosphorylation and Smad-dependent transcriptional activity [27,28]. To examine whether cortisol inhibits TGF-β signaling, HDFs were treated with cortisol in the presence of two different concentrations of cortisol (10 and 100 nM). The expression of TGF-β was decreased by cortisol in a dose-dependent manner (Figure 4A). To assess the effect of AP collagen peptides or GR inhibitors on TGF-β signaling, the supernatant of HDFs was analyzed to measure TGF-β levels (Figure 4B). Our results demonstrated that AP collagen peptides and the GR inhibitor recovered the cortisol-dependent decrease in TGF-β levels. To further investigate the effect of cortisol on TGF-β signaling, the phosphorylation of Smad 2/3 was examined using phosphor-Smad 2 or phosphor-Smad 2/3 antibodies.

Phosphorylation of Smad 2 and Smad 2/3 was inhibited by cortisol but recovered by AP collagen peptides or GR inhibitors (Figure 4C,D). Therefore, TGF-β levels are downregulated by cortisol, and AP collagen hydrolysates and GR inhibitors are effective in recovering the cortisol-mediated TGF-β signaling cascade.

### 2.5. AP Collagen Peptides Alleviate Cortisol-Dependent Inhibition of Collagen Type I in Replicative Senescent HDFs Presenting Increased Cortisol Sensitivity and GR Expression

To address the effect of cortisol on senescent cells, we produced replicative senescent HDFs and compared the expression of senescence-associated beta-galactosidase (SA-β-gal) in young HDFs (passage 5, p5) and senescent HDFs (passage 25, P25). Senescent HDFs showed morphological changes, such as a flat large cell shape and drastically elevated SA-β-gal activity compared to young HDFs (Figure 5A,B). The effects of AP collagen peptides on cortisol-simulated senescent HDFs were investigated. In senescent HDFs, cortisol treatment suppressed the production of procollagen type 1, and the inhibition was more susceptible to cortisol compared to young HDFs (Figure 5C). An approximate two-fold increase in the inhibition of procollagen type I was observed in senescent HDFs (data not shown). To explain the cortisol-susceptibility of senescent HDFs, we analyzed the transcript expression levels of GR (Figure 5D). At the cellular level, glucocorticoid sensitivity is closely influenced by the level of cellular GR expression [29,30]. Senescent HDFs expressed considerably higher levels of GR mRNA than young HDFs. Thus, we suggest that senescent HDFs are more susceptible to cortisol, presumably because of an increase in glucocorticoid receptor expression, and AP collagen peptides are effective in the prevention of the cortisol-dependent inhibition of collagen type I in senescent HDFs.

### 2.6. AP Collagen Peptides Prevent Cortisol-Dependent Inhibition of Collagen Synthesis in a Reconstituted Human Skin Model

Because our data demonstrated that AP collagen peptides had an inhibitory effect on cortisol-induced signaling and collagen type I loss in HDFs, we verified the effect of AP collagen peptides on collagen expression in a human reconstituted skin model by treatment with cortisol in the presence of AP collagen peptides, GR inhibitor or TGF-β, and collagen expression levels were determined. H&E staining and Masson’s trichrome staining showed that cortisol reduced the expression of collagen fiber, and AP collagen peptides, GR inhibitor and TGF-β reversed the inhibition of collagen fiber (Figure 6A). Immunostaining for collagen type I indicated that collagen type I was drastically decreased by cortisol, which was recovered by AP collagen peptides, GR inhibitor and TGF-β. mRNA analysis showed that cortisol treatment alleviated col1a1 mRNA expression, and the decrease in col1a1 mRNA was reversed by AP collagen peptides, GR inhibitor and TGF-β (Figure 6B). Finally, we analyzed the supernatants of the skin model and found that there was a recovery of cortisol-mediated procollagen type I suppression in an AP collagen peptide, GR inhibitor or TGF-β-treated skin model (Figure 6C). Collectively, the cortisol-induced suppression of collagen type I expression is recovered by AP collagen peptides in the dermis.

## 3. Discussion

Bioactive peptides (BPs) are defined as specific peptides that may have a positive effect on human health [31,32]. Bioactive peptides—such as the collagen hydrolysates obtained by the enzymatic hydrolysis of collagen-rich materials such as skin, bone and fish scales—are among the most used ingredients for the development of nutraceuticals. Consumption of collagen hydrolysates has been known to exert beneficial effects on joints [33], bones [33,34], skin [17,19] and blood vessels [35], which contain collagen. A metabolism study of collagen hydrolysate suggests that collagen-derived di- or tripeptides in the blood torrent increase significantly after oral ingestion and are incorporated into the skin, bone and joint tissue [36,37,38]. Hyp-containing peptide is resistant to digestion through the gastrointestinal tract and present in human plasma for a longer period than other collagen peptides, implying its physiological activities [39].

Collagen is the main ECM protein of connective tissues such as the skin, cartilage and blood vessels, providing structural integrity and performing various physiological functions [40,41]. Young skin is composed of 80% collagen type I, the main collagen in the skin, and 15% collagen type III [42]. Collagen production and the ratio of type I to type III gradually decline with age [43,44]. The loss of collagen production in the body starts from early adulthood and presents a 1–1.5% decrease per year [45,46]. Many anti-aging applications of a wide range of ligands, including antioxidants (vitamins, polyphenols and flavonoids) and cell regulators (retinol, peptides, hormones and botanicals) are targeted to stimulate the production of collagen and ECM components [47]. Collagen has been proven to be a potential dietary supplement for slowing down the effects of chronological skin aging [48,49,50,51] and skin photoaging [52,53,54]. One of the proposed functioning mechanisms of collagen hydrolysates is that collagen oligopeptides act as ligands and bind to receptors on the fibroblast membrane to stimulate the production of new collagen, elastin, and hyaluronic acid [55]. Generation of tri- and dipeptides was observed in the ear with dermatitis as a result of extensive degradation of ECM, indicating that collagen-derived tri- and dipeptides act as an ECM-associated signal to trigger fibroblast growth for tissue reconstruction [56]. In fact, Pro-Hyp was incorporated into fibroblasts with mesenchymal stem cell marker, p75NTR, indicating its role in wound healing [57]. Moreover, tri- and dipeptides containing Hyp induce chemotaxis of fibroblasts by acting on the membrane receptor [58,59] and, especially, Pro-Hyp stimulates cell proliferation and hyaluronic acid synthesis [60].

The magnitude of the cellular response to GC depends on the hormonal levels and its glucocorticoid sensitivity. The level of cortisol increases with aging [61,62], contributing to the development of aging phenotypes. In aged mouse skin, the elevated activity of 11β-hydroxysteroid dehydrogenase (HSD), an enzyme that catalyzes the conversion of inactive cortisone to active cortisol, may be linked to an increase in local GC synthesis and detrimental changes occurring in the skin with age [63]. Aged mice treated with 11β-HSD inhibitor or aged 11β-HSD KO mice exhibited protection against age-induced skin defects [64]. GC sensitivity is determined by the number of GRs and their ability to bind the GC-responsive element and/or other nuclear transcription factors to manipulate the expression of GC target genes [65,66]. Our results suggest that GR sensitivity and the number of GRs were increased in senescent HDFs compared to young HDFs. A study on the characterization of GR expression in human skin indicated that the amounts of GR were highest at the extremes of life and lowest at midlife in breast and abdominal skin [67]. Taken together, GR expression changes with aging, and the relationship between the number of glucocorticoid receptors and age, and qualitative changes in the quantity of receptor binding should be further determined.

In the epidermis, GR- and MR-mediated signaling act together to regulate epidermal development and inflammation [68]. However, the roles of GR and MR in the dermis have not been thoroughly investigated. Our present study suggests that GR and MR are co-expressed; GR may be expressed at higher levels with a Ct value of 25–26 compared to MR with a Ct value of 31–32 (data not shown). GR-induced signaling is more likely linked to cortisol-induced loss of collagen type I in human dermal fibroblasts. However, further studies should be conducted to compare the absolute expression levels of GR and MR. The inhibitory role of GR on TGF-β1 expression has been described in several cell types [69,70,71]. Our present study also demonstrated that GR inhibitors, as well as AP collagen peptides, prevented cortisol-induced activation of GR and a decrease in TGF-β signaling, indicating that AP collagen peptides could act as GR signaling inhibitors and effectively abolish the cortisol-induced GR signaling implicated in collagen loss. Thus, AP collagen peptides seem to be an effective bioactive supplement to rejuvenate damaged collagen, representing the visible signs of aging, such as fine lines, wrinkles and sagging.

## 4. Materials and Methods

### 4.1. Collagen Peptides and Chemicals

The AP collagen peptides used in this study were supplied by Aestura (Anseong, Korea) and produced by enzymatic degradation of collagen derived from the skin of golden threadfin bream (*Nemipterus virgatus*). AP collagen peptides contained at least 15% tripeptides Gly X–Y (X and Y are arbitrary but often occupied by proline, hydroxyproline, and alanine), including 3% Gly-Pro-Hyp (GPH). GPH was purchased from Bachem (Bubendort, Switzerland). Cortisol, GR inhibitors (RU486), MR inhibitors (spironolactone) and TGF-β were obtained from Sigma (St. Louis, MO, USA).

### 4.2. Cell Culture

Primary cultures of human dermal fibroblasts (HDFs) derived from human neonatal foreskins were purchased (Lonza, Basel, Switzerland) and maintained in Dulbecco’s modified Eagle’s medium (DMEM) supplemented with 10% (*v*/*v*) fetal bovine serum (FBS, Gibco, Brooklyn, NY, USA) and 1% penicillin-streptomycin solution (Gibco, Brooklyn, NY, USA) at 37 °C in a humidified incubator with 5% CO_2_. The confluent HDFs were transferred to two new dishes and cultured until they reached 90% confluence. HDFs that had been cultured for 5–6 passages were used for the experiments, except for the study on replicative senescent HDFs cultured for 25–26 passages.

### 4.3. Quantitative Real-Time RT-PCR (qRT-PCR)

For total RNA isolation, cells were harvested using the TRIzol reagent (Invitrogen, Carlsbad, CA, USA) and reconstituted skin models were homogenized using the RNeasy Mini kit (Qiagen, Hilden, Germany), according to the manufacturer’s instructions. The RNA concentration was determined spectrophotometrically and RNA integrity was assessed using a BioAnalyzer 2100 (Agilent Technologies, Santa Clara, CA, USA). Two micrograms of RNA were reverse-transcribed into cDNA using SuperScript III reverse transcriptase (Invitrogen) and aliquots were stored at −20 °C. TaqMan RT-PCR technology (7500Fast, Applied Biosystems, Foster City, CA, USA) was used to determine the expression levels of the selected target genes. The process included a denaturing step performed at 95 °C for 10 min; 50 cycles were performed at 95 °C for 15 s and 60 °C for 1 min. The following TaqMan probes were used for qRT-PCR analysis: Col1A1 (Hs00164804_m1) and GR (Hs00353740_m1). The probe for GAPDH (Hs02786624_g1) (Applied Biosystems) was also amplified to normalize variations in cDNA levels across different samples.

### 4.4. Western Blot Analysis

To prepare cell lysates, HDFs were washed with ice-cold PBS and lysed in RIPA buffer (50 mM Tris-HCl pH 7.4, 150 mM NaCl, 0.5% sodium deoxycholate, 0.1% SDS and 1% NP-40) in the presence of a protease and phosphatase inhibitor cocktail (Sigma). The lysates were then centrifuged at 15,000× *g* for 20 min and the supernatants were used for analysis. Protein concentrations were determined using a BCA kit (Sigma), with bovine serum albumin (BSA) as the standard. Equal amounts of protein (40 μg/well) from the cell lysates were loaded and separated using 8–12% gradient SDS-PAGE and transferred onto PVDF membranes. The membranes were blocked in 3% BSA in TBST (20 mM Tris-HCl, pH 8.5, 150 mM NaCl and 0.5% Tween) at room temperature for 30 min. The blots were incubated at 4 °C with anti-GAPDH (Sigma), anti-phospho-GR, anti-GR, anti-phospho-Smad 2, anti-Smad 2, anti-phospho-Smad 2/3 and anti-Smad 2/3 (Cell Signaling Technology, Danvers, MA, USA) antibodies overnight in 3% BSA in TBST. The membranes were washed three times for 15 min in TBST, followed by incubation with the appropriate horseradish peroxidase-conjugated goat anti-rabbit IgG secondary antibodies (Bio-Rad, Hercules, CA, USA) for 1 h at room temperature. The membranes were washed and visualized using an enhanced chemiluminescence reagent (ECL) for immunofluorescence staining (Amersham Pharmacia Biotech, Piscataway, NJ, USA). Image analysis of the immunoblots was performed using ImageQuant TL software (GE Healthcare Life Sciences, Pittsburgh, PA, USA).

### 4.5. ELISA

Media from HDFs were harvested and centrifuged for 15 min at 4 °C. The supernatants were freeze-dried and used for the measurement of procollagen type I C-peptide (Takara, Otsu, Japan) or TGF-β1 (R&D Systems, Minneapolis, MN, USA) using specific ELISA kits, according to the manufacturer’s instructions.

### 4.6. Senescence Associated Beta-Galactosidase (SA-β-Gal) Assay

SA-β-gal staining was performed using a β-gal staining kit (Invitrogen, Carlsbad, CA, USA), according to the manufacturer’s instructions. In brief, cells in culture were fixed using the SA-β-gal staining kit, and fixed cells were incubated with the staining solution prepared in a buffer (pH 6.0) overnight at 37 °C. Cells were imaged under a microscope (Olympus, Tokyo, Japan), and at least four images were randomly selected per sample. The percentage of SA-β-gal-positive cells was calculated by dividing the number of positive cells by the total number of cells.

### 4.7. Immunohistochemistry Analysis of a Skin Model

A reconstituted human epidermFT model (EFT-400) was purchased from MatTek Corp. (Ashland, MA, USA) and maintained according to the manufacturer’s instructions. The skin model was treated with cortisol in the presence of AP collagen peptides, GR inhibitor, or TGF-β. The culture media were analyzed for procollagen type I secretion using an ELISA kit, and the reconstituted human skin was freeze-dried for mRNA analysis or fixed in 10% neutral phosphate-buffered formalin and embedded in paraffin for special and immunohistochemical staining. Tissue sections (4 μm thick) were stained with hematoxylin and eosin (H&E) or Masson’s trichrome. Tissue sections were incubated with anti-collagen type I antibodies (Novus Biologicals, Centennial, CO, USA). HRP-conjugated donkey anti-rabbit IgG (H + L) antibodies (Novus, Centennial, CO, USA) were used as secondary antibodies. Immunoreactivity was visualized using 3,3-diaminobenzidine as a chromogen. The results were analyzed under a light microscope (BX53, Olympus, Tokyo, Japan) and photomicrographs were taken using a cooling digital camera (DP72, Olympus, Tokyo, Japan).

### 4.8. Statistical Analysis

The data were analyzed using Student’s *t*-test and expressed as the mean ± standard deviation (SD). Measurements were obtained for at least three independent experiments and representative results are shown.

## 5. Conclusions

In summary, the present study demonstrated that AP collagen peptides from fish skin could confer protection against the adverse effect of cortisol by increasing the synthesis of collagen type I in HDFs. The action mechanism underlying the beneficial effects of AP collagen peptides on the cortisol-mediated inhibition of collagen type I may be involved in blocking GR signaling, which is responsible for TGF-β activation. In replicate senescent HDFs presenting increased cortisol sensitivity, AP collagen peptides also effectively alleviated the cortisol-dependent inhibition of collagen type I. Finally, the effect of AP collagen peptides was proven to boost the synthesis of collagen type I in reconstituted human skin models. These results suggest that AP collagen peptides are potential dietary supplements for use against cortisol-related aging.

## Figures and Tables

**Figure 1 ijms-22-04788-f001:**
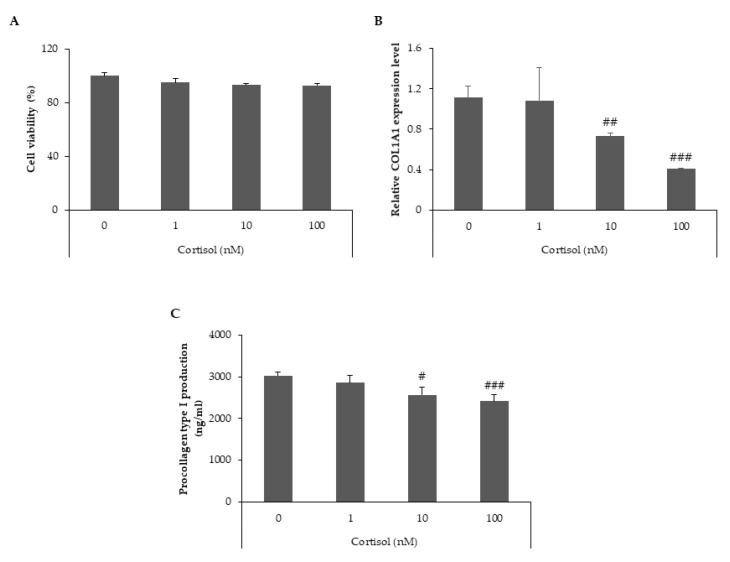
Cortisol downregulates the level of collagen type I in HDFs. HDFs were treated with different concentrations of cortisol (0, 1, 10 and 100 nM) and cultured for 24 h. (**A**) Cell viability was determined using the CCK-8 assay. (**B**) Total RNA was extracted from cells and relative col1a1 mRNA levels were measured by qRT-PCR. (**C**) Culture media were harvested and the levels of procollagen type I were determined using a specific ELISA kit. All data represent the mean ± SD of three independent experiments. Significant differences: # *p* < 0.05, ## *p* < 0.01, ### *p* < 0.005 compared with the control.

**Figure 2 ijms-22-04788-f002:**
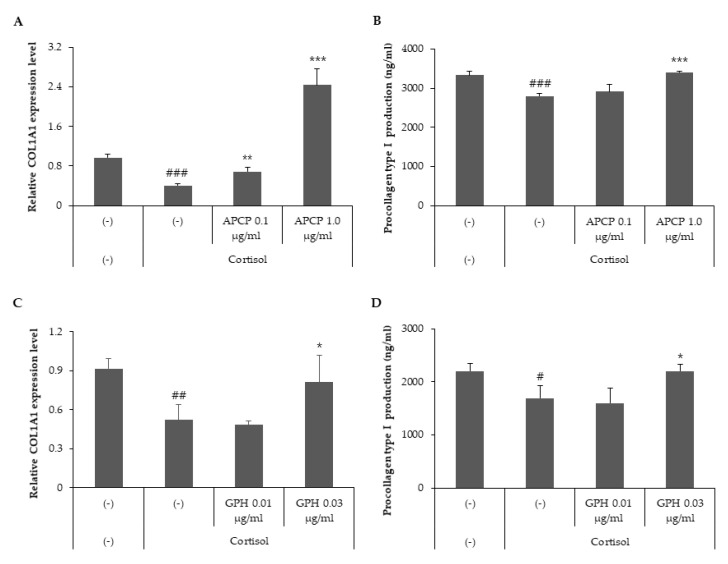
AP collagen peptides and their major component GPH prevent cortisol-dependent downregulation of collagen type I. HDFs were treated with cortisol (100 nM) in the presence of various concentrations of AP collagen peptides (APCP, 0, 0.1 and 1 μg/mL) or various concentrations of GPH (0, 0.01 and 0.03 μg/mL) and cultured for 24 h. (**A**,**C**) Total RNA was extracted from cells and relative col1a1 mRNA levels were measured by qRT-PCR. (**B**,**D**) Culture media were harvested and the level of procollagen type I was detected using a specific ELISA kit. All data represent the mean ± SD of three independent experiments. Significant differences: # *p* < 0.05, ## *p* < 0.01, ### *p* < 0.005 compared with the control; * *p* < 0.05, ** *p* < 0.01, *** *p* < 0.005 compared with cortisol-treated control.

**Figure 3 ijms-22-04788-f003:**
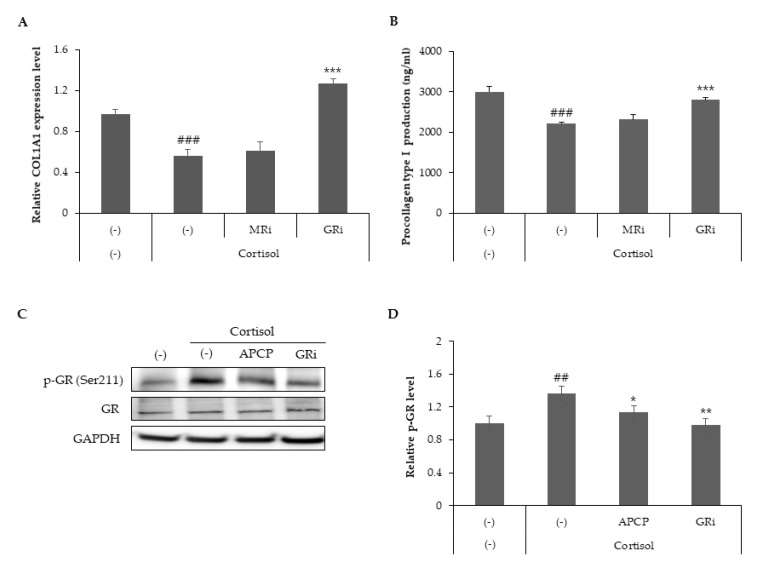
GR inhibitors restore cortisol-dependent inhibition of collagen type I synthesis and AP collagen peptides inhibit GR phosphorylation. HDFs were treated with cortisol (100 nM) in the presence of an MR inhibitor (spironolactone, 1 μM) or GR inhibitor (RU486, 1 μM) and cultured for 24 h. (**A**) Total RNA was extracted from cells and relative col1a1 mRNA levels were measured by qRT-PCR. (**B**) Culture media were harvested, and the level of procollagen type I was detected using a specific ELISA kit. (**C**,**D**) Cell lysates from cells cultured for 8 h were subjected to immunoblot analysis to assess the phosphorylation of GR. All data represent the mean ± SD of three independent experiments. Significant differences: ## *p* < 0.01, ### *p* < 0.005 compared with the control; * *p* < 0.05, ** *p* < 0.01, *** *p* < 0.005 compared with cortisol-treated control.

**Figure 4 ijms-22-04788-f004:**
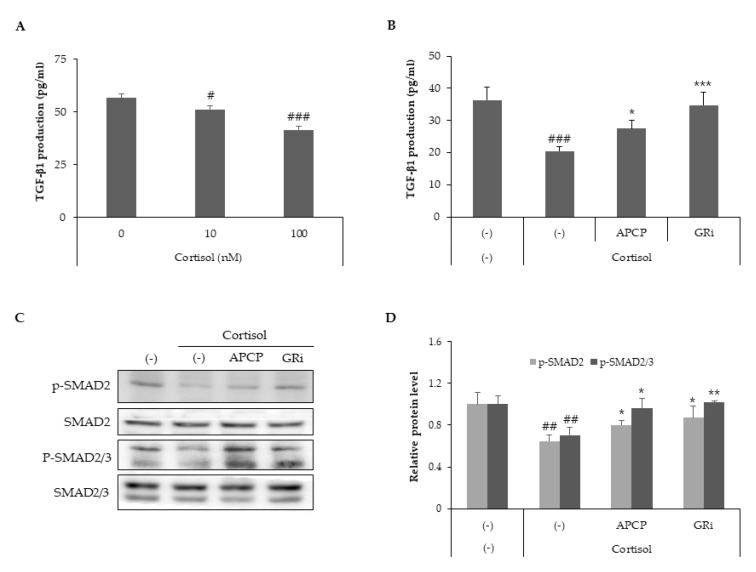
AP collagen peptides recover cortisol-dependent inhibition of TGF-β signaling. HDFs were treated with various concentrations of cortisol (0, 10 and 100 nM) and cultured for 24 h. (**A**) Culture media were harvested and TGF-β levels were determined using a specific ELISA kit. HDFs were treated with cortisol (100 nM) in the presence of APCP (1 μg/mL) or GR inhibitor (1 μM) and cultured for 8 h and 24 h. (**B**) Culture media from cells cultured for 24 h were harvested and TGF-β levels were detected using a specific ELISA kit. (**C**,**D**) Cell lysates from cells cultured for 8 h were subjected to immunoblot analysis to assess the phosphorylation of Smad2 (Ser465/Ser467) and Smad2 (Ser465/Ser467) / 3 (Ser423/425). All data represent the mean ± SD of three independent experiments. Significant differences: # *p* < 0.05, ## *p* < 0.01, ### *p* < 0.005 compared with the control; * *p* < 0.05, ** *p* < 0.01, *** *p* < 0.005 compared with cortisol-treated control.

**Figure 5 ijms-22-04788-f005:**
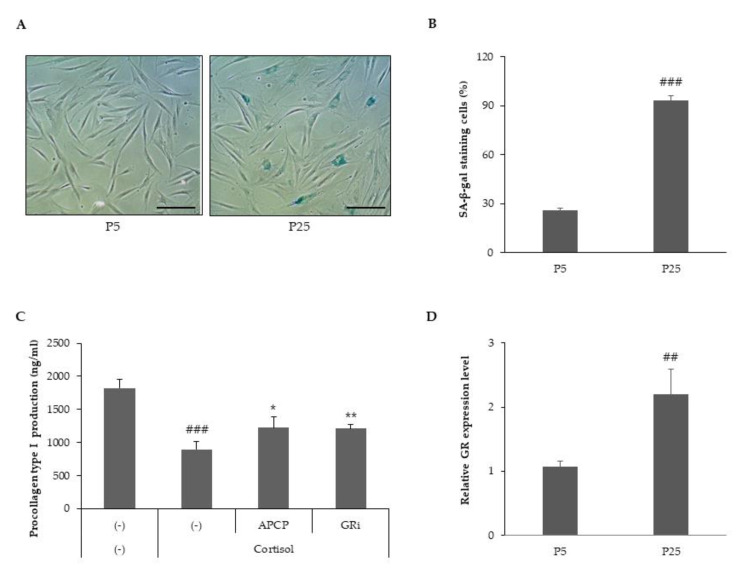
AP collagen peptides alleviate cortisol-dependent inhibition of procollagen type I in replicative senescent HDFs. (**A**) Young HDFs at passage 5 (P5) and replicative senescent HDFs at passage 25 (P25) were stained for the presence of senescence-associated beta-galactosidase (SA-β-gal) activity (blue). Scale bar = 200 μm. (**B**) The percentage of senescent cells was determined for SA-β-gal. Senescent HDFs were treated with cortisol (100 nM) in the presence of APCP (1 μg/mL) or GR inhibitor (1 μM) and cultured for 24 h. (**C**) Culture media from cells cultured for 24 h were harvested and the levels of procollagen type I were determined via a specific ELISA kit. (**D**) Total RNA was extracted from young HDFs and senescent HDFs, and relative GR mRNA was measured via qRT-PCR. All data represent the mean ± SD of three independent experiments. Significant differences: ## *p* < 0.01, ### *p* < 0.005 compared with control; * *p* < 0.05, ** *p* < 0.01 compared with cortisol-treated control.

**Figure 6 ijms-22-04788-f006:**
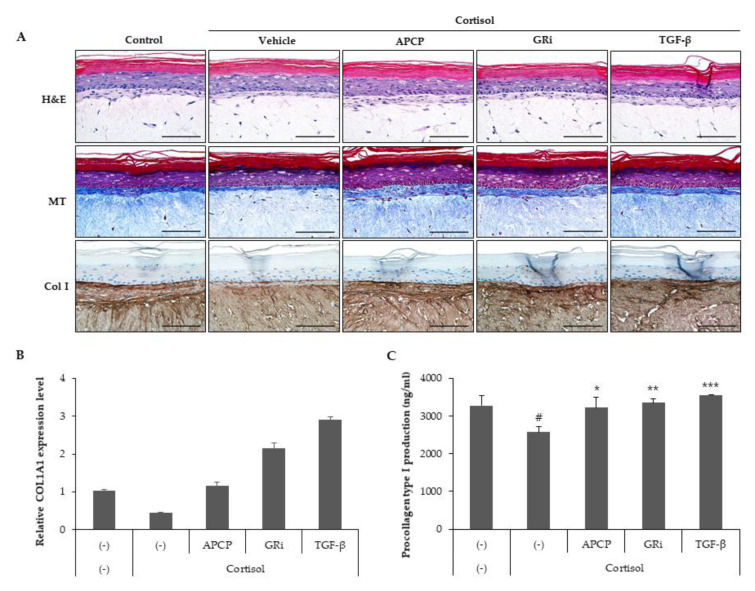
AP collagen peptides prevent cortisol-dependent inhibition of collagen synthesis in a reconstituted human skin model. The model was treated with cortisol (100 nM) in the presence of APCP (1 μg/mL), GR inhibitor (1 μM) or TGF-β (10 ng/mL) and cultured for 48 h. (**A**) Reconstituted human skin models were fixed and stained with H&E, Masson’s trichrome (MT) or immunohistochemical staining for collagen type 1 antibody. Scale bar: 100 μm. (**B**) Total RNA was extracted from a reconstituted human skin model, and relative col1a1 mRNA was measured via qRT-PCR. Data represent the mean ± SD of two independent experiments. (**C**) Secreted procollagen type 1 was detected in culture supernatants using a specific ELISA kit. Data represent the mean ± SD of three independent experiments. Significant differences: # *p* < 0.05 compared with control; * *p* < 0.05, ** *p* < 0.01, *** *p* < 0.005 compared with cortisol-treated control.

## Data Availability

Not applicable.

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
