# Peer review of "AP Collagen Peptides Prevent Cortisol-Induced Decrease of Collagen Type I in Human Dermal Fibroblasts"

_ijms, 2021, doi:10.3390/ijms22094788_

Round 1

Reviewer 1 Report

The article fits the aims of the journal and falls within the journal purpose. The authors aim to assess the effects of collagen peptides produced by enzymatic degradation of collagen derived from fish skin on fibroblast production of type I collagen, and on the molecular mechanistic links in its production.

The research is sound, adds elements of novelty; the manuscript is fairly documented and the methodology is adequately chosen. Data are adequately analysed, results are presented in a clear manner. All figures are illustrative for the results and statistically significant differences are readily visible from the included images.The entire section is well constructed and well written, data are presented in the proper form, statistics are well employed, sustaining the further conclusions, that seem sufficiently well supported by results. The article may be an useful contribution to the journal; however few changes should be taken into consideration:

  1. authors are advised to define all abbreviations at first appearance in text; in abstract and also elsewhere, including but not limted to ‘AP collagen’.
  2. Defining AP collagen in Abstract section as well defining and as further ellaborating on Introduction, not only in Methodology, would be highly beneficial to the reader.
  3. Grammar and punctuation must also be carefully checked within the entire article (e.g. title: col-Lagen, etc). References style must be uniform and consistent throughout the entire article (e.g. line 228 skin 17, 19 and blood vessels {Zhang, 2010 #1998 which contain collagen).
  4. Discussion section needs to be expanded, and the mechanisms of collagen oligopeptides acting as ligands binding to fibroblast surface receptors might be further ellaborated on.
  5. A proper separate Conclusion section must be inserted, in the benefit of the reader. Currently, the conclusions are mixed with discussion and do not stand out.

Author Response

1. authors are advised to define all abbreviations at first appearance in text; in abstract and also elsewhere, including but not limted to ‘AP collagen’.

: Thank you for your comment. As you point out, there were several abbreviations missing definition such as transforming growth factor (TGF)-β (line 17, 59), extracellular matrix (ECM) (line 49), Glycine-Proline- Hydroxyproline (Gly-Pro-Hyp, GPH) (line 12, 52), and we correct it. AP collagen is the raw material name of collagen supplied from Aestura (Korea) and contains no abbreviation.

2. Defining AP collagen in Abstract section as well defining and as further ellaborating on Introduction, not only in Methodology, would be highly beneficial to the reader.

: I elaborated AP collagen in Abstract (line 11) and Introduction (line 61) for better understanding to the readers.

3. Grammar and punctuation must also be carefully checked within the entire article (e.g. title: col-Lagen, etc). References style must be uniform and consistent throughout the entire article (e.g. line 228 skin 17, 19 and blood vessels {Zhang, 2010 #1998 which contain collagen).

: Thank you for your comment. We carefully checked within the entire article and correct wrong grammar and punctuation. We also made reference style consistent.

4. Discussion section needs to be expanded, and the mechanisms of collagen oligopeptides acting as ligands binding to fibroblast surface receptors might be further ellaborated on.

: We discussed more on the mechanisms of collagen (line235-2237, 251-257). Thank you for your comment.

5. A proper separate Conclusion section must be inserted, in the benefit of the reader. Currently, the conclusions are mixed with discussion and do not stand out.

: We have added separate conclusion following materials and methods. Thank you.

Reviewer 2 Report

This article addresses a topic well known to the dermatologist. in particular, it confirms the aspects relating to the pathogenesis of skin atrophy induced by chronic treatments with corticosteroids. The role of AP collagen peptides is very interesting and deserves further experimentation through dedicated clinical studies, also in order to evaluate their possible practical use.

Author Response

Thank you for your kind and generous review. Hopfully, we  further conduct clinical studies with AP collagen peptides.

Round 2

Reviewer 1 Report

The authors have modified the manuscript according to previous suggestions; the manuscript is now significantly improved;

I consider it could be published in the journal in this current form.